# The Ethanolic Extract of *Gomphrena celosioides* Mart. Does Not Alter Reproductive Performance or Embryo-Fetal Development, nor Does It Cause Chromosomal Damage

**DOI:** 10.3390/pharmaceutics14112369

**Published:** 2022-11-03

**Authors:** Fabricia Rodrigues Salustriano, Antonio Carlos Duenhas Monreal, Silvia Cordeiro das Neves, Giovana Martins de Oliveira, Diego Duarte Marques de Oliveira, Marcelo Luiz Brandão Vilela, Valter Aragão do Nascimento, Allana Cristina Faustino Martins, Baby Saroja, Arunachalam Karuppusamy, Henrique Rodrigues Scherer Coelho, Candida Aparecida Leite Kassuya, Dayanna Isabel Araque Gelves, Marcos José Salvador, Rodrigo Juliano Oliveira, Roberto da Silva Gomes

**Affiliations:** 1Centro de Estudos em Células Tronco, Terapia Celular e Genética Toxicológica (CeTroGen), Campo Grande 79070-900, MS, Brazil; 2Faculdade de Medicina Dr. Hélio Mandetta (FAMED), Universidade Federal de Mato Grosso do Sul (UFMS), Campo Grande 79070-900, MS, Brazil; 3Campus II de Três Lagoas, Universidade Federal de Mato Grosso do Sul, Três Lagoas 79602-070, MS, Brazil; 4Programa de Pós-graduação em Saúde e Desenvolvimento da Região Centro-Oeste, Campo Grande 79070-900, MS, Brazil; 5Department of Pharmaceutical Sciences, North Dakota State University, Fargo, ND 58102, USA; 6Faculdade de Ciências da Saúde, Universidade Federal da Grande Dourados (UFGD), Dourados 79804-970, MS, Brazil; 7Instituto de Biologia, Departamento de Biologia Vegetal, Universidade Estadual de Campinas (UNICAMP), Campinas 13083-862, SP, Brazil

**Keywords:** *Gomphrena celosioides*, chromosomal damage, toxicity, teratogenesis

## Abstract

*Gomphrena celosioides* is a native Brazilian plant found in the State of Mato Grosso do Sul. It is used in folk medicine to treat kidney diseases, skin diseases, infections, rheumatism, gastrointestinal diseases, and respiratory diseases. It is also used as an abortifacient. To evaluate the effects of the ethanolic extract of *Gomphrena celosioides* (EEGc) on reproductive performance, embryo development, and chromosome stability, Swiss mice were randomly divided into experimental groups (*n* = 10). The animals in the control group received the vehicle Tween 80–1% in the proportion of 0.1 mL/10 g of body weight orally, from the first to the 18th gestational day. The animals in the treatment groups received the EEGc (100, 1000, and 2000 mg/kg) from the first to the 18th gestational day. The animals underwent evaluations of their reproductive performance and embryofetal development. The results showed that the EEGc did not change the animals’ final weight, weight gain, uterine weight, or net weight gain. The evaluation showed that the absolute and relative organs’ weights did not vary between the different experimental groups. In addition, the EEGc did not change the numbers of implants, live fetuses, dead fetuses, or fetal resorptions. There were no differences in post-operative loss rates, implantations, or resorptions, nor were there differences in fetal viability or sex ratio. The use of the EEGc did not result in different frequencies of malformations. In addition, the EEGc did not alter the frequency of chromosomal damage or frequency of micronuclei. Based on our findings, we considered the extract of *Gomphrena celosioides* to be safe for use during pregnancy, although some parameters indicated caution in its use.

## 1. Introduction

*Gomphrena celosioides* Mart. Is a weed that belongs to the *Amarahthaceae family*. It is popularly known as “perpétua-brava”, “perpétua”, “bachelor’s button”, and “prostrate globe-amaranth” [1,2]. This species has a confirmed occurrence in the following regions of Brazil: the north region (Pará, Rondônia, Roraima, and Tocantins); the northeast region (Bahia, Maranhão, Pernambuco, and Rio Grande do Norte); the midwest region (Distrito Federal, Goiás, Mato Grosso do Sul, and Mato Grosso); the southeast region (Espírito Santo, Minas Gerais, Rio de Janeiro, and São Paulo); and the southern region (Paraná, Rio Grande do Sul, and Santa Catarina). In addition, there is a possible occurrence in the states of Amazonas, Amapá, Alagoas, Ceará, Paraíba, and Piauí e Sergipe. Its presence has been confirmed in the Amazon, Caatinga, Cerrado, and the Atlantic Forest [3,4]. In addition to its presence in Brazil, this species is found throughout South America, Asia, and Africa [2,5,6].

*G. celosioides* is popularly used for the treatment of urinary tract disorders and kidney stones [2,7,8,9], gastrointestinal and respiratory diseases [9], rheumatism [10], infections [11] and as a natural analgesic [10]. People in South America continue to use this plant as an abortifacient [12]. In Asia, this plant is used to treat liver disorders [6,13]. 

We previously demonstrated that the oral administration of the ethanolic extract of *Gomphrena celosioides* (EEGc) in mice significantly increased urine production, dependent on nitric oxide, prostaglandin, and bradykinin pathways. After seven days of treatment, the effect was sustained and coincided with the reduction in serum aldosterone [9]. We also demonstrated the maintenance of the diuretic effect of the EEGc in renovascular hypertensive rats, with reduced blood pressure after the first week of treatment by inhibiting the angiotensin-converting enzyme. These effects were long-lasting and strong enough to prevent cardiac remodeling and, therefore, have potential as an antihypertensive [2].

*G. celoisioides* also possesses scientifically confirmed anti-arthritic, antihyperalgesic [10], gastroprotective [14], antimicrobial [11,15], anti-inflammatory [16], and antioxidant [17] activity. Given the above, the EEGc has potential for the development of diuretic, natriuretic, and/or antihypertensive drugs. 

Due to its effects as described above, this plant can be prescribed for pregnant women [18]. However, not much information is known about the toxicological potential of *G. celosioides*. According to Macorini et al. [10], a single dose of 2000 mg/kg did not induce clinical signs of toxicity in Wistar rats, which suggested that the lethal does (LD50) is higher. Regarding subacute toxicity, repeated doses showed that 75, 150, and 300 mg/kg did not induce adverse clinical signs or damage to target tissues. According to these authors, and according to the Globally Harmonized System of classification, the EEGc dosages can be placed in Category 5, which is least toxic or non-toxic. No reports of toxicity in pregnant females were found in the literature consulted. 

Accordingly, and mainly because this plant is reported to be used as an abortifacient in popular medicine in South America [12], it is necessary to evaluate the effects of the EEGc on reproductive performance, embryo-fetal development, and chromosomal stability.

## 2. Materials and Methods

### 2.1. Plant Material of G. celosioides Extract

Aerial parts of *G. celosioides* were collected in April 2014 in Paranaíba, Mato Grosso do Sul, Brazil [lat: −19.666667 long: −51.183333 WGS84]. The plant material was identified by Professor Dr. Josafá Carlos de Siqueira from Pontifícia Universidade Católica do Rio de Janeiro (PUC-RJ), and the voucher specimen was deposited in the herbarium of the Faculty of Philosophy, Sciences, and Letters of Ribeirão Preto, Universidade de São Paulo (FFCLRP/USP) under number SPFR-2962. This species was registered on the National System for the Management of Genetic Heritage and Associated Traditional Knowledge (SISGEN) under codes AF99A04 and ASEC80E.

The ethanolic extract of *G. celosoides* (EEGc) was prepared, and its chemical composition was analyzed by mass spectrometry, following the method described by Vasconcelos et al. [9].

### 2.2. Animals

Sixty mice (*Mus musculus*) of Swiss lineage of both sexes were used. They were of reproductive age, with 40 females with an average weight of 33 g and 20 males with an average weight of 35 g. All of the animals underwent an adaptation period of seven days and were placed in mini-isolators (*Rack* ventilada Alesco^®^, MS, Brazil) lined with wood shavings *Pinus* sp. Males were kept alone and females were kept in pairs by mini-isolator. The animals were fed with standard commercial feed (Nuvital^®^, MS, Brazil) and filtered water under a free-access system. The light was controlled by photoperiod (12 h light/12 h dark), the temperature was kept at 22 ± 2 °C, and the humidity was 55 ± 10. The research was carried out according to the protocols of the Universal Declaration of Animal Rights and approved by the Ethics Committee on the Use of Animals (CEUA) of Universidade Federal de Mato Grosso do Sul (UFMS), under number 965/2018.

After we simultaneously evaluated teratogenesis and genotoxicity, mice were chosen as experimental models. The spleen of mice is less efficient at sequestering micronucleated cells, compared with rats and humans [19]. Thus, quantifying micronucleated cells in rat blood, for example, could underestimate the frequency of lesions. Therefore, mice became the most suitable experimental model.

### 2.3. Experimental Design

The animals were mated overnight in the ratio of two females to one male. Pregnancy detection was performed by observing the vaginal plug, which indicated the zero-day of pregnancy [20,21,22,23,24,25]. The pregnant animals were randomly distributed among the experimental groups (*n* = 10), In the control group, the animals received the vehicle (Tween 80–1%) in the proportion of 0.1 mL/10 g of body weight (b.w.), orally (p.o.), from the first to the 18th gestational day (g.d.). The treatment group of animals received the EEGc (100, 1000, and 2000 mg/kg, b.w./p.o.) from the first to the 18th g.d. The 100 mg/kg dose was chosen from the effective dose described by Vasconcelos et al. [9]. From this dose, the dose of 10× higher (1000 mg/kg) than that with the desired activity [26,27] was administered, with modifications.The limit dose of 2000 mg/kg was defined according to the acute oral toxicity guidelines [28].The *n* of at least 10 pregnant animals per group was based on OECD [26].

The same animals were used in the teratogenicity and genotoxicity tests. However, there was no positive control for these two assays simultaneously that could still follow the same proposed experimental design, i.e., being administered for 18 consecutive days. Thus, we chose not to use a positive control group for this study, as this could have negatively impacted the treatment schedule. Moreover, this study was delineated to follow the philosophy of the 3Rs (replacement, reduction, and refinement) proposed by Russell and Burch [29]. Another fact that prevented us from using a positive control group was that products known to be teratogenic, such as cyclophosphamide (20 mg/kg) [20], would have led to the description of a large number of malformations in the tables, which would make it difficult to understand the results. This would happen because the magnitude of the malformations found in positive controls would in no way resemble what was induced by a medicinal plant with a low capacity to induce teratogenesis.

### 2.4. Biological Tests

Peripheral blood was collected via puncturing the caudal vein on the 16th, 17th, and 18th g.d. On the 18th g.d., the animals were euthanized, followed by laparotomy, hysterectomy, omphalectomy, thoracotomy, weighing, and, eventually, storage of organs (lung, heart, spleen, liver, kidneys, and placenta) and fetuses.

The fetuses were collected and weighed and underwent systematic analysis of external malformations and sexing. Then, they were randomly assigned to two subgroups, each with approximately 50% of the offspring. The fetuses of the first group were fixed in Bodian’s solution ((distilled water (142 mL), acetic acid (50 mL), formaldehyde (50 mL), and 95% alcohol (758 mL)) for at least seven days and destined for visceral analysis [30] by means of microdissection with strategic cuts, to enable us to study the chest and abdomen, as proposed by Barrow and Taylor [31], and to study of head, according to Wilson [31], subject to the changes proposed by Oliveira et al. [32], Damasceno et al. [33], and Oliveira et al. [20]. The classification of visceral changes was based on the works of Taylor [34], Manson and Kang [35], Wise et al. [36], Damasceno et al. [33], and Oliveira et al. [20]. 

The fetuses of the second subgroup were fixed in absolute acetone for at least seven days and subjected to skeletal analysis by the technique described by Staples and Schnell [37], as modified by Oliveira et al. [20]. After fixation, the fetuses were eviscerated and immersed in KOH solution (0.8%) for the diaphanization process. Then, four drops of Alizarin Red were added. This solution was changed every 24 h for four consecutive days. After the fetuses were stained, the KOH solution was replaced by a whitening solution (1 L of glycerin, 1 L of ethyl alcohol, and 0.5 L of benzyl alcohol) and changed every 24 h for five consecutive days. Classifications of malformations were based on studies by Taylor [34], Manson et al. [35], Wise et al. [36], Damasceno et al. [33], and Oliveira et al. [20].

The fetal viscera and skeletons were analyzed using a stereomicroscopic loupe (Nikon^®^—SMZ 745T, MS, Brazil) with four-fold magnification.

### 2.5. Biometric Parameters

The calculations of biometric parameters consisted of the data on initial weight (females weighed on day zero), final weight (females weighed on the 18th g.d.), weight gain (final weight to initial weight), the weight of the uterus, and net weight gain (weight gain—uterus weight), in addition to the absolute and relative weight data (organ weight/final weight) of the heart, lung, spleen, kidneys, and liver.

### 2.6. Reproductive Performance and Embryo-Fetal Development

Reproductive parameters related to the fertility rate (number of pregnant females × 100/number of mated females); the number of implants as quantified and/or calculated (number of live fetuses + number of dead fetuses + number of resorption); the number of live fetuses; fetal viability (number of live fetuses × 100/number of implants); the rate of post-implantation losses ((number of implants − number of live fetuses) × 100/number of implants); the number of resorptions; the resorption rate (number of resorptions × 100/number of implants); placental weight; fetal weight; the placental index (placental weight/fetal weight); placental efficiency (fetal weight/placental weight); and sex ratio (number of male fetuses × 100/number of female fetuses) [23].

The classification of fetal weight according to gestational age (CPFIG) was performed in two ways: (1) according to Soulimane-Mokhtari et al. [38], fetuses were considered suitable for gestational age (SUGA) when they did not differ by more than ±1.7 × standard deviation (SD) from the mean of the control fetuses; they were considered small for gestational age (SGA) when their body weight was less than −1.7 × standard deviation in relation to the average of the fetuses in the control group; and they were considered large for gestational age (LGA) when they had a body weight greater than +1.7 × standard deviation in relation to the mean of the control group; (2) according to Oliveira et al. [20], the fetuses were classified as having an appropriate weight for pregnancy age (AWPA) when the weight of the fetus was between the average weight of the fetuses in the control group, more or less the standard deviation; fetuses were classified as having low weight for pregnancy age (LWPA) when the weight of the fetus was below the average weight of the fetuses in the control group minus the standard deviation of this same group; and fetuses were classified as having high weight for pregnancy age (HWPA) when the weight of the fetus was greater than the average weight of the fetuses in the control group plus the standard deviation of this same group. Soulimane-Mokhtari et al. [38] classified the fetus weights individually, and Oliveira et al. [20] classified the offspring in general.

### 2.7. Micronucleus in Peripheral Blood

The technique used for the trial development was based on the proposal by Hayashi et al. [19] and modified by Carvalho et al. [39]. By puncturing the caudal vein, 20 μL of peripheral blood was collected, deposited on a slide previously stained with Acridine Orange (1 mg/mL), and covered with a coverslip. The material remained stored in a freezer at −20 °C for a minimum period of seven days and was evaluated under a fluorescence microscope (Zeiss^®^, MS, Brazil) at 40× magnification, with a 420 nm to −490 nm excitation filter and a 520 nm barrier filter. Two thousand reticulocytes/slides/times were analyzed, totaling 6000 per animal.

### 2.8. Splenic Phagocytosis

The splenic phagocytosis assay was performed according to the procedure described by Schneider et al. [40]. The spleen was macerated in a physiological solution to obtain a homogeneous suspension of cells, and 100 μL of the cell suspension was placed on a slide previously stained with Acridine Orange (1 mg/mL), covered by a coverslip and stored in a freezer at −20 °C until the time of analysis. The analysis was performed under a fluorescence microscope (Zeiss^®^) at 40× magnification, with a 420 nm to 490 nm filter and a 520 nm barrier filter. One hundred cells per animal were analyzed. The absence or presence of phagocytosis was based on the description by Hayashi et al. [19] and Carvalho et al. [39].

### 2.9. Statistical Analysis

The ANOVA/Tukey test and the Chi-square test were used to compare the frequencies between the groups. The qualitative data and the frequencies obtained, as recommended by the specialized literature, used the offspring as the base unit [41]. The data were presented as mean ± standard error of the mean or mean ± standard deviation and the level of significance established was *p <* 0.05.

## 3. Results

### 3.1. Biometric Parameters

The animals from the different experimental groups started the experiment with similar weights (*p* > 0.05). The EEGc did not alter the final weight, weight gain, uterine weight, or net weight gain (*p* > 0.05). The evaluation of the absolute and relative weights of the heart, lung, spleen, liver, and kidneys did not vary between the different experimental groups (*p* > 0.05) (Table 1).

### 3.2. Reproductive Performance

The EEGc treatment did not change the numbers of implants, live fetuses, dead fetuses, or resorptions (*p* > 0.05). There were also no differences in the rates of fertility, post-implantation losses, resorption, fetal viability, or the sex ratio (*p* > 0.05) (Table 2).

### 3.3. Embryofetal Development and Placental Efficiency

The EEGc treatment did not change fetal weight, placental weight, or the placental index (*p* > 0.05). However, there was a statistically significant reduction in placental efficiency (*p* < 0.05) in the groups treated with the lowest and highest EEGc dose (Table 3).

When assessing the adequacy of weight to gestational age, it was observed that 92.80% of the fetuses in the control group were classified within the range of 0.95 to 1.39 (which refers to the mean weight of the control ±1.7 × 0.13 (standard deviation of the control weight)). The percentages of weight adequacy to the gestational age did not vary significantly in the groups treated with the EEGc, according to Soulimane-Mokhtari et al. [38]. Regarding the adequacy of weight to gestational age, according to Oliveira et al. [20], the offspring were considered to have appropriate weights for gestational age (AWGA) (Table 3).

The systematic analysis of external malformations indicated the occurrence of limb hyperflexion (uni and bilateral), micromelia, inadequate paw rotation, hyperextension of the anterior and posterior limbs, folded tail, and paw hyperflexion. The total frequency of malformations did not differ between the control group and those treated with the EEGc (*p* > 0.05) (Table 4).

Regarding visceral malformations, hydrocephalus (mild and moderate degrees), hydronephrosis, and choanal obstruction were identified. There were no statistically significant differences between the control group and those treated with the EEGc (*p* > 0.05) (Table 5).

The skeletal malformations found were reduced ossification (OR) of the frontal, parietal, palate, presfenoid, supraoccipital, external center, rib, manubrium, xiphoid process, phalanx, metacarpal and metatarsus; rib, phalanx, and metacarpal agenesis; bifurcated rib; abnormal sternum; and poorly positioned fibula. The EEGc treatment increased (*p* < 0.05) the frequency of abnormal sternum and fibula rotation over the tibia (Table 6).

### 3.4. Toxicogenetics

The frequency of chromosomal damage did not vary between different experimental groups, which indicated that the EEGc did not change the frequency of micronuclei (*p* > 0.05). There were also no significant variations between the different analysis times, which indicated that the EEGc had no cumulative effect (*p* > 0.05) (Figure 1A).

The frequency of splenic phagocytosis was similar among all experimental groups (*p* > 0.05), indicating that the EEGc did not cause an increase in phagocytic activity (Figure 1B).

## 4. Discussion 

The consulted literature indicates that there are no data available on the teratogenic and genotoxic effects of *G. celosioides*, which demonstrates the need for and the pioneering nature of this study, mainly due to its ethnopharmacological indications [2,7,8,9,11]. It is noteworthy that, despite its ethnopharmacological indications, *G. celosioides* is described as an abortifacient [12]. This fact reinforced the need for that study, which characterized the reproductive performance of females.

It Is widely known that pregnant women should expose themselves as little as possible to medications during pregnancy [42], due to the risk of miscarriage [43] and malformations [44]. However, there are situations in which pregnant women need to be medicated.

Hypertension is a disease that affects 5% to 10% of pregnant women, and the incidence increases with obesity, pregnancy at an advanced age, and comorbidities due to previous diseases, such as diabetes mellitus and kidney diseases [45,46].

Hypertension during pregnancy has an established treatment [47]. However, developing safe drugs is always required, as not all patients respond equally and adequately to therapies [48]. Previous studies by our group demonstrated that the extract of *G. celosioides* has a diuretic, natriuretic potential [9], and antihypertensive effects [2]. Accordingly, we evaluated the EEGc in a model with teratogenesis and chromosomal stability.

Our results suggest that daily doses of up to 2000 mg/kg are not maternotoxic, do not alter the reproductive performance or chromosomal stability of pregnant females, and do not alter embryo-fetal development.

The absence of maternal toxicity is indicated by the lack of changes in the biometric parameters and by the absolute and relative weight of the organs. Changes in these parameters, especially weight loss, are associated with toxic effects on exposures during pregnancy [49,50]. This reinforces the absence of toxicity of the EEGc and the low frequency of genotoxic damage evidenced by the micronucleus assay. Chromosomal lesions, in general, are considered an important toxicity biomarker [25,30,51].

The EEGc changed no parameters related to reproductive performance. These results suggest that the extract does not alter nesting and does not cause resorption and/or post-implantation losses, and it does not alter fetal viability or sex ratio. These results suggest that females and pregnant women can safely consume the EEGc. Therefore, these data allow us to infer that *G. celosioides*, according to the present experimental design, is not abortive, as the female animals were treated during the entire gestational period, including the pre-implantation period (from the first to the fifth d.g.), and did not present gestational loss. This fact can be stated because there were no differences in fertility rates. Furthermore, no reduction in the number of offspring was observed. Therefore, these results do not support the use of this plant as an abortifacient, as Burkill [12] reported.

Another fact that merits attention is that the extract did not alter embryo-fetal development. However, it reduced placental efficiency in the groups that were treated with the lowest and highest dose of EEGc. Despite this reduction in placental efficiency, there was no variation in fetal weight, placental weight, the placental index, the percentage of fetuses with adequate weight for the age of the pregnancy, or the adequacy of the weight of the offspring to the age of the pregnancy. These results, again, indicate the safety of using the EEGc during the gestational period.

The reduction in placental efficiency, which is an effect of the treatment, did not determine variation in the adequacy of weight for the age of pregnant women or the adequacy of weight to gestational age. Placental efficiency is defined as the weight (in grams) of the fetus produced by the weight (in grams) of the placenta [52,53,54,55]. This parameter, which varies between species, reflects the maternal–fetal relationship established during the gestational period and is a determinant of intrauterine growth, as it is responsible for the fetus’s nutritional and hormonal supply [54]. Therefore, placental efficiency is influenced by the size, morphology, blood flow, and transport efficiency that can occur through simple diffusion and/or active transport [53,56,57]. In addition, the placenta synthesizes and metabolizes key nutrients for fetal growth and hormonal supply [57,58]. Therefore, any change in the homeostasis of these processes can lead to embryophetolality and/or embryophetotoxicity, delayed growth, delayed development, and teratogenesis. Although this change did not occur for the EEGc, it is not uncommon for other extracts, such as those of *Gochnatia polymorpha* and *Croton urucurana* [55,59].

Another interesting fact that this study brings to the literature is the adjustment of weight to gestational age, as proposed by Soulimane-Mokhtari et al. [38]. The calculation is made for each fetus, and the adequacy of weight to the age of pregnancy as proposed by Oliveira et al. [20], which is calculated for the offspring, showed no differences, suggesting that both the classification of the fetus and the classification of offspring yield similar results in interpreting the effects of a product during the gestational period. However, new studies are needed to compare these two methodologies to confirm this conclusion.

The absence of maternal toxicity is indicated by the lack of changes in the absolute and relative weights that are among the biometric parameters for the organs. Changes in these parameters, especially changes in weight reduction, are associated with toxic effects due to exposures during pregnancy [23,55,60]. This reinforces the absence of toxicity of the EEGc and the low frequency of genotoxic damage evidenced by the micronucleus assay.

Further, this reinforces the absence of genetic toxicity and the low frequency of phagocytic activity. In general, splenic phagocytosis occurs when cells with DNA lesions are circulating [51]. Thus, for example, the spleen functions as a filtering network capable of sequestering micronucleated cells [51]. As these two biomarkers appeared in low frequency in this study, it is assumed that the EEGc does not cause damage to the DNA and, therefore, does not stimulate an increase of the phagocytic activity of the spleen.

The frequency of chromosomal damage observed, even for the highest dose of EECc, was within what the literature recommends as basal damage. According to Ishikawa et al. [24], the average frequency of micronuclei for the negative control, treated with 1% Tween 80 (the same diluent and the same concentration that were used in this study) can reach 7.1 ± 1.03 micronuclei. Furthermore, Vani et al. [25,30,61] reported a basal frequency of micronuclei for pregnant female Swiss mice of up to 8.90 ± 1.35, 5.20 ± 1.31, and 8.90 ± 1.35.

Regarding malformations, it was found that the groups treated with the two lowest doses of the EEGc showed an increase in the incidence of unilateral limb hyperflexion, and only the lowest dose showed a rise in fetuses with sternums showing abnormal ossification and rotation of the fibula over the tibia. These malformations are not severe, would not prevent quality survival, and could regress until the end of pregnancy. If they were present in humans, the fibula rotation could be corrected surgically and/or with physiotherapy. The same occurrence was observed in the control group for the other visceral and skeletal changes, which allowed us to infer that these were variants of normality. In general, variants of normality are changes that may regress with advancing gestation or at birth, as the fetuses in this study were collected at the 18th g.d. The birth would only happen after 21 days. Variants of normality are described in several articles in the area [23,24,25,30].

Another fact to be highlighted is that the malformations that presented statistically significant differences did not present a dose–response relationship; that is, what was expected was that with the increase in the EEGc dose, there would also be an increase in the frequency or severity of malformations. However, this was not observed. Thus, the effect may not be caused by the administration of the EEGc. In this case, the hypothesis is that these are, in fact, variants of normality.

Other experimental designs are available in the literature and can complement the data produced by this study, which is novel and pioneer in presenting the effects of *G. celosioides* on maternal performance and embryofetal development. The specific protocols of the OECD—Test No. 421 (2016), OECD—Test No. 414 (2017), and OECD—Test 443 (2018) guidelines should be considered in future studies.

## 5. Conclusions

Based on our findings, we consider that the ethanolic extract *G. celosioides* (EEGc) is safe for use during pregnancy, although the results indicated that caution should be exercised in its use. Considering the cost-benefit ratio, the extract demonstrated safe use and an indication for therapeutic use during pregnancy. Furthermore, the extract did not show an abortifacient effect.

## Figures and Tables

**Figure 1 pharmaceutics-14-02369-f001:**
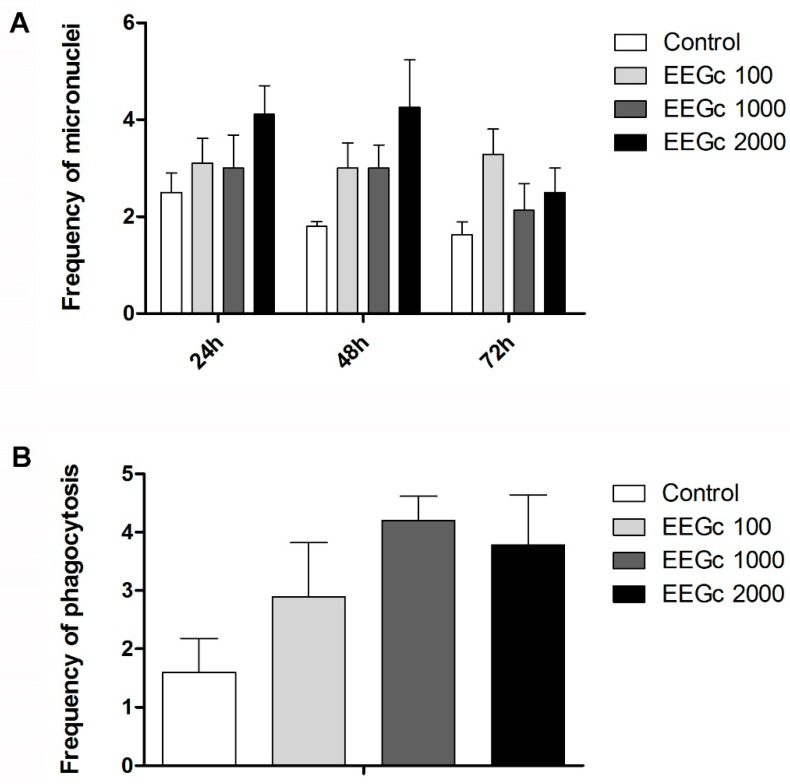
(**A**) Frequency of micronuclei in females treated with the ethanolic extract *G. celosioides* (EEGc). (**B**) Frequency of splenic phagocytosis in females treated with the ethanolic extract *G. celosioides* (EEGc). Statistical test: ANOVA/Tukey (*p* > 0.05).

**Table 1 pharmaceutics-14-02369-t001:** Biometric parameters, absolute weight, and relative weight of the organs of females treated with the ethanolic extract of *G. celosioides* (EEGc).

Parameters	Experimental Groups			
	Control	EEGc 100	EEGc 1000	EEGc 2000
Initial weight (g)	34.05 ± 1.95 ^a^	31.89 ± 0.98 ^a^	33.31 ± 1.17 ^a^	32.66 ± 0.90 ^a^
Final weight (g)	52.19 ± 1.71 ^a^	49.41 ± 2.49 ^a^	54.02 ± 2.06 ^a^	51.33 ± 1.45 ^a^
Weight gain (g)	18.14 ± 1.69 ^a^	17.52 ± 2.83 ^a^	20.71 ± 1.67 ^a^	18.67 ± 1.24 ^a^
Uterus weight (g)	18.19 ± 0.67 ^a^	16.27 ± 1.65 ^a^	18.23 ± 1.35 ^a^	17.59 ± 0.98 ^a^
Net weight (g)	−0.052 ± 1.47 ^a^	1.25 ± 1.35 ^a^	2.47 ± 0.64 ^a^	1.07 ± 0.69 ^a^
Absolute Weight (g)				
Heart	0.18 ± 0.0099 ^a^	0.19 ± 0.0083 ^a^	0.20 ± 0.0133 ^a^	0.17 ± 0.0047 ^a^
Lung	0.25 ± 0.0119 ^a^	0.24 ± 0.0123 ^a^	0.26 ± 0.0104 ^a^	0.25 ± 0.0087 ^a^
Spleen	0.15 ± 0.0129 ^a^	0.14 ± 0.0102 ^a^	0.15 ± 0.0085 ^a^	0.14 ± 0.1077 ^a^
Liver	2.14 ± 0.0801 ^a^	2.25 ± 0.0938 ^a^	2.29 ± 0.0843 ^a^	2.14 ± 0.0739 ^a^
Kidneys	0.45 ± 0.0293 ^a^	0.38 ± 0.0169 ^a^	0.40 ± 0.0094 ^a^	0.40 ± 0.0151 ^a^
Relative Weight				
Heart	0.0034 ± 0.0002 ^a^	0.0039 ± 0.0002 ^a^	0.0036 ± 0.0002 ^a^	0.0034 ± 0.0000 ^a^
Lung	0.0049 ± 0.0002 ^a^	0.0049 ± 0.0003 ^a^	0.0049 ± 0.0003 ^a^	0.0049 ± 0.0002 ^a^
Spleen	0.0029 ± 0.0002 ^a^	0.0028 ± 0.0001 ^a^	0.0028 ± 0.0002 ^a^	0.0027 ± 0.0002 ^a^
Liver	0.041 ± 0.0023 ^a^	0.046 ± 0.0018 ^a^	0.043 ± 0.0017 ^a^	0.042 ± 0.0015 ^a^
Kidneys	0.0087 ± 0.0004 ^a^	0.0078 ± 0.0004 ^a^	0.0074 ± 0.0003 ^a^	0.0079 ± 0.0003 ^a^

Legend: Equal letters indicate the absence of statistically significant differences; EEGc—ethanolic extract *G. celosioides*. Mean + standard error of the mean (Anova/Tukey test; *p* > 0.05).

**Table 2 pharmaceutics-14-02369-t002:** Reproductive parameters of females treated with the *G. celosioides* ethanolic extract (EEGc).

Parameters	Experimental Groups (mg/kg)			
	Control	EEGc 100	EEGc 1000	EEGc 2000
Mating	10	10	10	10
Fertility rate (%)	100	100	100	100
Implants	13.10 ± 0.78 ^a^	12.30 ± 1.22 ^a^	14.60 ± 0.98 ^a^	13.20 ± 0.87 ^a^
Live fetuses	12.50 ± 0.69 ^a^	11.50 ± 1.25 ^a^	12.70 ± 1.06 ^a^	12.30 ± 0.73 ^a^
Dead fetuses	0.00 ± 0.00 ^a^	0.00 ± 0.00 ^a^	0.10 ± 0.10 ^a^	0.00 ± 0.00 ^a^
Resorption	0.60 ± 0.27 ^a^	0.90 ± 0.38 ^a^	1.80 ± 0.76 ^a^	0.90 ± 0.50 ^a^
PILR (%)	4.14 ± 1.94 ^a^	7.45 ± 2.87 ^a^	12.65 ± 5.43 ^a^	6.24 ± 2.93 ^a^
RR (%)	4.14 ± 1.94 ^a^	8.11 ± 3.09 ^a^	12.09 ± 5.35 ^a^	6.24 ± 2.93 ^a^
Fetal viability (%)	95.85 ± 1.94 ^a^	92.55 ± 2.87 ^a^	87.35 ± 5.43 ^a^	93.76 ± 2.93 ^a^
Sexual reason	2.36 ± 1.00 ^a^	1.68 ± 0.46 ^a^	1.41 ± 0.31 ^a^	1.36 ± 0.27 ^a^

Legend: PILR—post-implantation loss rate; RR—resorption rate; EEGc—ethanolic extract *G. celosioides*. Equal letters indicate the absence of statistically significant differences. Mean + standard error of the mean (ANOVA/Tukey test; *p* > 0.05).

**Table 3 pharmaceutics-14-02369-t003:** Embryo–fetal development parameters and placental efficiency of females treated with the *G. celosioides* (EEGc) ethanolic extract.

Parameters	Experimental Groups (mg/kg)			
	Control	EEGc 100	EEGc 1000	EEGc 2000
Fetal weight ^1,^*	1.17 ± 0.13 ^a^	1.18 ± 0.16 ^a^	1.17 ± 0.12 ^a^	1.20 ± 0.009 ^a^
Placenta weight ^1,#^	0.07 ± 0.006 ^a^	0.07 ± 0.002 ^a^	0.07 ± 0.002 ^a^	0.08 ± 0.007 ^a^
Placental index ^1,#^	0.06 ± 0.005 ^a^	0.06 ± 0.001 ^a^	0.06 ± 0.001 ^a^	0.07 ± 0.005 ^a^
Placental efficiency ^1,#^	19.6 ± 0.59 ^c^	17.23 ± 0.43 ^a,b^	18.51 ± 0.44 ^b,c^	15.99 ± 0.35 ^a^
%SGA ^2^	4.80	6.14	2.40	1.63
%SUGA ^2^	92.80	86.84	95.20	98.37
%LGA ^2^	2.40	7.02	2.40	0.00
AJWPA—Oliveira		AWPA	AWPA	AWPA

Legend: CFWGA [38]—classification of fetal weight according to gestational age; SUGA—suitable for gestational age; SGA—small for gestational age; LGA—large for gestational age; AJWPA [20]—adjustment of weight of pregnancy age; AWPA—fetuses with appropriate weight pregnancy age; EEGc—ethanolic extract *G. celosioides*. Different letters indicate statistically significant differences (statistical test: ^1^ YEAR/Tukey; *p* < 0.05). * Mean ± standard deviation. # Mean ± standard error of the mean. ^2^ Chi-square test (Control × EEGc) (*p* > 0.05).

**Table 4 pharmaceutics-14-02369-t004:** External malformations found in the offspring of females treated with the *G. celosioides* ethanolic extract (EEGc).

Parameters	Experimental Groups (mg/kg)			
	Control	EEGc 100	EEGc 1000	EEGc 2000
External Analysis				
Analyzed fetuses	125	113	126	106
Normal fetuses	121	94	114	98
%	96.80	83.19	90.48	92.45
Bilateral limb hyperflexion	2	0	0	1
%	0.80	0.00	0.00	0.94
Unilateral limb hyperflexion	1	7 *	10 *	3
%	0.80	6.09	7.87	2.83
Hyperkyphosis	0	3	0	0
%	0.00	2.61	0.00	0.00
Micromelia	0	3	1	1
%	0.00	2.61	0.79	0.94
Poorly rotating paw	1	6	0	0
%	0.80	5.22	0.00	0.00
Hind limb hyperextension	0	4	2	4
%	0.00	3.48	1.57	3.77
Anterior limb hyperextension	0	0	0	1
%	0.00	0.00	0.00	0.94
Limb hyperextension	0	4	2	5 *
%	0.00	3.48	1.57	4.72
Folded tail	0	1	0	0
%	0.00	0.87	0.00	0.00
Paw hyperflexion	0	1	0	0
%	0.00	0.87	0.00	0.00
Frequency of external malformation	4	19	12	8
% external malformation	3.20	16.81	9.52	7.55

Legend: % = percentage of fetuses with malformation; EEGc—ethanolic extract *G. celosioides*. * Statistically significant difference in relation to the control group (statistical test: Chi-square; *p* < 0.05).

**Table 5 pharmaceutics-14-02369-t005:** Visceral malformations found in the offspring of females treated with the *G. celosioides* ethanolic extract (EEGc).

Parameters	Experimental Groups (mg/kg)			
	Control	EEGc 100	EEGc 1000	EEGc 2000
Visceral Analysis				
Analyzed fetuses	62	54	61	59
Normal fetuses	22	26	19	19
%	35.48	48.15	31.15	32.2
Mild hydrocephalus	36	19	36	38
%	58.06	35.19	60	64.41
Moderate hydrocephalus	0	0	1	0
%	0	0	1.67	0
Hidronefrose	11	13	16	5
%	17.74	24.07	26.67	8.47
Obstructed choana	1	0	0	0
%	1.61	0	0	0
Frequency of visceral malformation	40	28	42	40
% visceral malformation	64.52	51.85	70.00	67.8

Legend: % = percentage of fetuses with malformation; EEGc—ethanolic extract *G. celosioides*. (Statistical test: Chi-square; *p* > 0.05).

**Table 6 pharmaceutics-14-02369-t006:** Skeletal malformations found in the offspring of females treated with the *G. celosioides* ethanolic extract (EEGc).

Parameters	Experimental Groups (mg/kg)			
	Control	EEGc 100	EEGc 1000	EEGc 2000
Skeletal Analysis				
Analyzed fetuses	63	59	65	47
Normal fetuses	7	3	2	8
%	11.11	5.18	3.08	17.02
Skull				
Front OR	4	7	8	2
%	6.35	11.86	12.50	4.26
OR Interparietal	0	1	0	1
%	0.00	1.69	0.00	2.13
OR Palate	6	5	7	3
%	9.52	8.47	10.94	6.38
OR Presfenoid	7	15	9	5
%	11.11	25.42	14.06	10.64
OR Supra occipital	0	1	0	0
%	0.00	1.69	0.00	0.00
Body				
OR External center	43	45	54	27
%	68.25	76.27	84.38	57.45
Rib agenesis	0	1	0	0
%	0.00	1.69	0.00	0.00
Bifurcated rib	0	1	0	0
%	0.00	1.69	0.00	0.00
OR ribs	0	0	0	1
%	0.00	0.00	0.00	2.13
Abnormal sternebrio	1	8 *	3	1
%	1.59	13.56	9.69	2.13
OR Manubrium	0	0	1	0
%	0.00	0.00	1.56	0.00
OR Xiphoid process	7	15	2	0
%	11.11	25.42	3.13	0.00
Limb				
Poorly positioned fibula	0	2	1	3
%	0.00	3.39	1.56	6.38
Phalange agnesia	13	8	12	3
%	20.63	13.56	18.75	6.38
OR Phalanx	28	19	19	23
%	44.44	32.20	29.69	48.94
Metacarpal agenesis	0	1	0	0
%	0.00	1.69	0.00	0.00
OR Metacarpal	2	0	1	0
%	3.17	0.00	1.56	0.00
OR Metatarsus	1	0	0	0
%	1.59	0.00	0.00	0.00
Rotation of the fibula over the tibia	2	11 *	5	6
%	3.17	18.64	7.81	12.77
Frequency of visceral malformation	56	56	63	39
% visceral malformation	88.89	94.91	96.92	82.98

Legend: % = percentage of fetuses with malformation; RO—reduced ossification; EEGc—ethanolic extract *G. celosioides*. * Statistically significant difference (statistical test: Chi-square; *p* < 0.05).

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
