# Peer review of "The Ethanolic Extract of Gomphrena celosioides Mart. Does Not Alter Reproductive Performance or Embryo-Fetal Development, nor Does It Cause Chromosomal Damage"

_pharmaceutics, 2022, doi:10.3390/pharmaceutics14112369_

Round 1
Reviewer 1 Report
This study aimed to ascertain whether the ethanolic extract of Gomphrena celosioides (EEGc), a plant used in folk medicine for treatment of various pathological conditions, was able to affect reproductive performance and embryofetal development. In this respect, mice received by oral route EEGc at the doses of 100, 1000 and 2000 mg/kg b.wt. from the 1st to the 18th gestational day.
Introduction well resumes a series of effects of Gomphrena celosioides (Gc), described in the literature, and also reports original results previously obtained by the Authors with EEGc including its effects on diuresis, natriuresis and blood pressure homeostasis through actions on several pathways such as the nitric oxide, prostaglandin, bradykinin and rennin-angiotensin-aldosterone ones. However, considering that Gc has a reputation of being an abortifacient in popular medicine of South America, Authors rightly believed to be necessary to clarify the effects of EEGc on reproductive performance, embryofetal development and chromosomal stability.
Materials and Methods report with meticolousness and competence the adopted experimental protocols in the EEGc-treated mice. The performed tests included the collection of organs and fetuses, and the calculation of biometric parameters. Reproductive performance and embryofetal development were assessed through numerous parameters such as the fertility rate, number of implants and live fetuses, fetal viability and weight, and so on. Micronuclei formation was verified in peripheral blood and splenic phagocytosis was also studied by a suitable assay.
Results clearly report the obtained evidences according to the above protocols. Six tables and one figure well represent such evidences.
Discussion underlines how the present results appear to exclude for EEGc toxic gestacional (teratogenic) effects involving reproductive performance, embryofetal development and chromosomal stability. The observed reduction in placental efficiency by EEGc represented an isolated datum that, by itself, was not correlated with the other studied parameters. Therefore, the same results seem to give the lie to the use of Gc as an abortifacient. It was concluded that EEGc is safe during pregnancy although its employment needs adequate caution.
Overall, this study is somewhat original, well planned and correctly performed on a methodological point of view. The manuscript has been prepared with accuracy notwithstanding several sentence redundancies and some minor lexical imprecisions needing, here and there, easy adjustments (e.g. line 335: …an isolated data=…an isolated datum). Therefore, if possible, Authors might try to speed up and synthesize, where advisable, their phrasing to enhance the efficacy of the manuscript and the value of the obtained results.
Author Response
Response to reviewer 1:
This study aimed to ascertain whether the ethanolic extract of Gomphrena celosioides (EEGc), a plant used in folk medicine for treatment of various pathological conditions, was able to affect reproductive performance and embryofetal development. In this respect, mice received by oral route EEGc at the doses of 100, 1000 and 2000 mg/kg b.wt. from the 1st to the 18th gestational day.
Introduction well resumes a series of effects of Gomphrena celosioides (Gc), described in the literature, and also reports original results previously obtained by the Authors with EEGc including its effects on diuresis, natriuresis and blood pressure homeostasis through actions on several pathways such as the nitric oxide, prostaglandin, bradykinin and rennin-angiotensin-aldosterone ones. However, considering that Gc has a reputation of being an abortifacient in popular medicine of South America, Authors rightly believed to be necessary to clarify the effects of EEGc on reproductive performance, embryofetal development and chromosomal stability.
Materials and Methods report with meticolousness and competence the adopted experimental protocols in the EEGc-treated mice. The performed tests included the collection of organs and fetuses, and the calculation of biometric parameters. Reproductive performance and embryofetal development were assessed through numerous parameters such as the fertility rate, number of implants and live fetuses, fetal viability and weight, and so on. Micronuclei formation was verified in peripheral blood and splenic phagocytosis was also studied by a suitable assay.
Results clearly report the obtained evidences according to the above protocols. Six tables and one figure well represent such evidences.
Discussion underlines how the present results appear to exclude for EEGc toxic gestacional (teratogenic) effects involving reproductive performance, embryofetal development and chromosomal stability. The observed reduction in placental efficiency by EEGc represented an isolated datum that, by itself, was not correlated with the other studied parameters. Therefore, the same results seem to give the lie to the use of Gc as an abortifacient. It was concluded that EEGc is safe during pregnancy although its employment needs adequate caution.
Overall, this study is somewhat original, well planned and correctly performed on a methodological point of view. The manuscript has been prepared with accuracy notwithstanding several sentence redundancies and some minor lexical imprecisions needing, here and there, easy adjustments (e.g. line 335: …an isolated data=…an isolated datum). Therefore, if possible, Authors might try to speed up and synthesize, where advisable, their phrasing to enhance the efficacy of the manuscript and the value of the obtained results.
Response:
We thank the reviewer for the comment. The manuscript was reviewed, and typos and redundancies were corrected (please verify the highlighted changes in the main manuscript).
Thus, I think I have complied with the points raised by both reviewers.
I hope that the revised manuscript can be published as is without further delay.
With all good wishes,
Roberto Gomes

Reviewer 2 Report
This paper reports for the first time an assessment of the potential developmental toxicity and genotoxicity of an ethanolic extract of Gomphrena celosioides Mart. The results suggest that this extract does not cause developmental toxicity in mice, and does not induce micronucleus formation. However, there are some limitations to the methodology used as well as the English language in some places.
Some of these limitations are noted below and it is suggested that these be addressed in the paper.
The authors state that they have examined effects on reproductive performance and embryo-fetal development, however reference to reproductive performance is slightly misleading as the study more closely resembles a developmental toxicity test than a reproductive toxicity study. The test item was only administered during gestation and the dams were killed prior to delivery. There was no administration to males prior to or during mating, and no administration to dams prior to or during mating, or during the post-natal period.
Mice have been used in the present study whereas rats tend to be the preferred rodent species for developmental and reproductive toxicity studies. It is suggested that text should be added to indicate why the test species was selected and is appropriate. Discussion on why the number of animals per group (n=10) is appropriate would also be useful, noting that the OECD test guideline for developmental toxicity (TG 414) recommends at least 16 animals with implantation sites.
Another limitation of the study is the absence of a positive control for the micronucleus assay which is needed to confirm the ability of the test system to detect substances that induce chromosomal damage. Reference is also not made to historical vehicle control data for the laboratory which would support the conclusion that the higher frequency of micronuclei at the high dose compared with controls (Figure 1A) is not suggestive of a positive response. This should be discussed.
Some more specific comments are below.
Abstract
The English language used in the abstract needs to be clarified. For example:
Lines 28-30: 'Given the above, to evaluate the effects of ethanolic extract of EEGc on reproductive performance, embryo development and chromosome stability.' Could change to something along the lines of 'Given the above, a study was undertaken to evaluate...'
Lines 35-37: 'The evaluation of the absolute and relative organ weights did not vary between the different experimental groups, it also did not change the number of implants...' It is not the evaluation that didn't change, it is the test item that didn't affect these parameters. Could possibly reword to say that there were no significant differences in these compared with controls..
Lines 39-40: 'The frequency of chromosomal damage did not change by EEGc in the frequency of micronuclei.' Could change to e.g. 'the frequency of chromosomal damage following treatment with EEGc was similar to controls', or was not affected by treatment.
Introduction
It would be helpful to be clearer about how much information is available on the safety of EEGc to date, or that there are few data on this. A recent paper on safety is not included and could be cited here and/or in the discussion: Preclinical safety evaluation of the ethanolic extract from the aerial parts of Gomphrena celosioides Mart. in rodents - PubMed (nih.gov)
Materials and Methods
Lines 121-123: the OECD references to support the high dose selected are not appropriate: neither of these studies (acute toxicity study and reproduction/developmental toxicity screening study) were conducted here. Further, the limit dose suggested in OECD TG 421 is 1000 mg/kg bw/day, not 2000 mg/kg bw/day.
Section 2.7: more clarity is needed on how the micronucleus test was performed: were the blood samples collected from the pregnant females or a satellite group? Were all 10/group examined? What cells were scored for micronuclei? OECD TG 474 recommends scoring at least 4000 immature erythrocytes. What were the criteria for acceptability of the test and for determining a positive or negative response? One would expect reference to historical control data as well as the use of a positive control to demonstrate validity of the test system.
Statistics: the appropriate statistical unit for evaluating developmental toxicity studies is the litter. It is unclear if this was used for this study.
Results
Lines 228-229: refers to a significant decrease in placental efficiency at the low and high dose. Lines 266-267: refers to increased frequency of abnormal sternum and rotation of the fibula at the low dose only. However, a clear dose-response was not observed for any of these changes, have the authors considered that these might be random statistical differences rather than treatment-related? Reference to historical control data would be helpful to interpret these results.
Lines 248-250: it is stated that treatment had no effect on the frequency of malformations (Table 4). However, several significant differences are shown in Table 4. Again these may not be treatment-related as they did not show a clear dose-response.
Discussion
Reference is made to changes in placental efficiency and malformations, but as noted above more consideration should be given as to whether these are treatment-related or due to chance.
Discussion of limitations of the study would be useful, e.g. absence of a positive control for the micronucleus assay, no reference to historical control data etc.
Discussion of what further research might be needed to confirm the safety of medicinal use of EEGc would be useful.
Author Response
Response to reviewer 2:
This paper reports for the first time an assessment of the potential developmental toxicity and genotoxicity of an ethanolic extract of Gomphrena celosioides Mart. The results suggest that this extract does not cause developmental toxicity in mice, and does not induce micronucleus formation. However, there are some limitations to the methodology used as well as the English language in some places.
Some of these limitations are noted below and it is suggested that these be addressed in the paper.
- The authors state that they have examined effects on reproductive performance and embryo-fetal development, however reference to reproductive performance is slightly misleading as the study more closely resembles a developmental toxicity test than a reproductive toxicity study. The test item was only administered during gestation and the dams were killed prior to delivery. There was no administration to males prior to or during mating, and no administration to dams prior to or during mating, or during the post-natal period.
Response:
We thank the reviewer's comments. We agree that there are different protocols that can be used for reproductive toxicity testing whether they focus on males or females. These designs include, but are not limited to, testing when the test drug (or extract) is administered to males before or during mating or to mothers before or during mating or during the postnatal period. All these designs are important and are very well described in the literature. However, these designs are not the object of study of our research. Our main objective, which was successfully accomplished, was to evaluate the effect of the extract on reproductive performance and embryofetal development when the treatment was performed after mating and throughout the gestational period. This design makes an important contribution to the field of reproductive toxicology and teratogenesis, and it can be seen in scientific papers published in high impact scientific journals. We highlighted some of these papers from our group that used exactly this design (Vani et al., 2021; Ishikawa et al., 2018; Vani et al., 2018a; Vani et al., 2018b) and also from other research groups that used designs similar in rats (Leal-Silva et al., 2022; Carvalho et al., 2021; Paula et al., 2020). These facts demonstrate that the selected experimental model is robust and contributes significantly to the production of knowledge. Therefore, we request that the reviewer consider publishing the article in its present form. We further point out that our inferences and conclusions were made based on the results we produced. We did not extrapolate the data to discuss any situation outside the proposed design. Thus, we understand that it is possible to talk about the absence of alteration in reproductive performance. However, we are willing to change to maternal performance which may be a less comprehensive term if it is of interest to the journal.
Carvalho, M., Caixeta, G. A. B., Lima, A. R. S., Teófilo, M. N. G., de Melo Cruvinel, W., Gomes, C. M., ... & Amaral, V. C. S. (2021). Assessing the safety of using the dry extract of Justicia pectoralis Jacq.(Acanthaceae) during pregnancy of Wistar rats. Journal of Ethnopharmacology, 268, 113618.
Ishikawa, R. B., Vani, J. M., das Neves, S. C., Rabacow, A. P. M., Kassuya, C. A. L., Croda, J., ... & Oliveira, R. J. (2018). The safe use of Doliocarpus dentatus in the gestational period: Absence of changes in maternal reproductive performance, embryo-fetal development and DNA integrity. Journal of ethnopharmacology, 217, 1-6.
Leal-Silva, T., Souza, M. R., Cruz, L. L., Moraes-Souza, R. Q., Paula, V. G., Soares, T. S., ... & Volpato, G. T. (2022). Toxicological effects of the Morinda citrifolia L. fruit extract on maternal reproduction and fetal development in rats. Drug and Chemical Toxicology, 1-7.
Paula, V. G., Cruz, L. L., Sene, L. B., Gratão, T. B., Soares, T. S., Moraes-Souza, R. Q., ... & Volpato, G. T. (2020). Maternal-fetal repercussions of Phyllanthus niruri L. treatment during rat pregnancy. Journal of ethnopharmacology, 254, 112728.
Vani, J. M., de Carvalho Schweich-Adami, L., Auharek, S. A., Antoniolli-Silva, A. C. M. B., & Oliveira, R. J. (2021). Pyriproxyfen does not cause microcephaly or malformations in a preclinical mammalian model. Environmental Science and Pollution Research, 28(4), 4585-4593.
Vani, J. M., de Carvalho Schweich, L., de Oliveira, K. R. W., Auharek, S. A., Cunha-Laura, A. L., Antoniolli-Silva, A. C. M. B., ... & Oliveira, R. J. (2018a). Evaluation of the effects of the larvicides temephos on reproductive performance, embryofetal development and DNA integrity of Swiss mice. Pesticide biochemistry and physiology, 148, 22-27.
Vani, J. M., Monreal, M. T. F. D., Auharek, S. A., Cunha-Laura, A. L., de Arruda, E. J., Lima, A. R., ... & Oliveira, R. J. (2018b). The mixture of cashew nut shell liquid and castor oil results in an efficient larvicide against Aedes aegypti that does not alter embryo-fetal development, reproductive performance or DNA integrity. Plos one, 13(3), e0193509.
Mice have been used in the present study whereas rats tend to be the preferred rodent species for developmental and reproductive toxicity studies. It is suggested that text should be added to indicate why the test species was selected and is appropriate. Discussion on why the number of animals per group (n=10) is appropriate would also be useful, noting that the OECD test guideline for developmental toxicity (TG 414) recommends at least 16 animals with implantation sites.
Response:
We thank the reviewer's comment. The literature does not contraindicate the use of mice. A fact that demonstrates this is that important scientific vehicles are publishing articles on teratogenesis made in mice (Stoev, 2022; Hu et al., 2021; Vani et al., 2021; Ishikawa et al., 2018; Vani et al., 2018a; Vani et al., 2018b; Scheller; Kalmring; Schubert, 2016; Koch; Noldner; Leuschner, 2013). In our case, we chose to use mice because (I) we have a limited amount of extract and using mice would increase by about 10x the amount of extract needed for the study since mice weigh about 10x more than mice; and (II) we generally associate the micronucleus test with teratogenesis assays.
Mouse spleens are less efficient at sequestering micronucleated cells than rat and human spleens (Hayashi et al., 1990). Therefore, quantifying these micronucleated cells in mice may yield an underestimated value. Thus, we always chose to perform the test in mice as it would be the best model to assess chromosomal damage and teratogenesis at the same time.
The paragraph (…)
Once we evaluated teratogenesis and genotoxicity simultaneously, mice were chosen as an experimental model. The spleen of mice is less efficient at sequestering micronucleated cells compared to rats and humans (Hayashi et al., 1990). Thus, quantifying micronucleated cells in rat blood, for example, could underestimate the frequency of lesions. Therefore, mice becomes the most suitable experimental model.
(…) was inserted in item 2.2. animals.
The sample n = 10 animals/group was chosen following guidance from the Ethics Committee on the Use of Animals (CEUA) of the Federal University of Mato Grosso do Sul, which emphasizes the 3Rs Principle. In the work “The principles of humane experimental technique” (Russel; Burch, 1959) the PRINCIPLE OF THE 3 R's of animal experimentation emerged, which are: (I) “replace”, which translates as replacing sentient animals, that is, capable of to experience pain, pleasure, happiness, fear, frustration and anxiety; (II) “reduction”, which means reducing the number of animals used, without impairing the reliability of the results; and (III) “refinement”, which means the decrease in the incidence or severity of applied procedures. In previous studies, we asked the CEUA for a greater number of animals to meet the directions of the guidelines. However, we were asked about the 3Rs Principle and, in a joint study with this commission, we could see that the literature has numerous articles published with experimental groups formed by 10 animals and this does not affect the statistical analysis (Lu et al., 2021; Vani et al., 2021; Bueno et al., 2020; Kamali et al., 2020; Saeed et al., 2020; Ishikawa et al., 2018; Vani et al., 2018a; Vani et al., 2018b ; Saini et al., 2013). Since then, we have used 10 animals/experimental group and our statistical analyzes remain robust. We also clarify that we are aware of the recommendations of the OECD test guideline for developmental toxicity (TG 414) recommends at least 16 animals with implantation sites. However, this same guideline reports that (…) “This Guideline for developmental toxicity testing is designed to provide general information concerning the effects of prenatal exposure on the pregnant test animal and on the developing organism; this may include assessment of maternal effects as well as death, structural abnormalities, or altered growth in the fetus. Functional deficits, although an important part of development, are not a part of this Guideline. They may be tested for in a separate study or as an adjunct to this study using the Guideline for developmental neurotoxicity. For information on testing for functional deficiencies and other postnatal effects, the Guidelines 416, 421/422, 426 and 443 (3-7) should be consulted” (we request the reviewer to pay attention to the item highlighted in yellow).
We rely on OECD 421 which reports that (…) “It is recommended that each group be started with at least 10 males and 12-13 females. Females will be evaluated pre-exposure for oestrous cyclicity and animals that fail to exhibit typical 4-5 day cycles will not be included in the study; therefore, extra females are recommended in order to yield 10 females per group. Except in the case of marked toxic effects, it is expected that this will provide at least 8 pregnant females per group which normally is the minimum acceptable number of pregnant females per group. The objective is to produce enough pregnancies and offspring to assure a meaningful evaluation of the potential of the chemical test to affect fertility, pregnancy, maternal and suckling behavior, and growth and development of the F1 offspring from conception to day 13 postpartum” (we ask the reviewer to pay attention to the item highlighted in yellow). We have reached the minimum number of females.
As requested by the reviewer, the sentence (…)
“The n of at least 10 pregnant animals/group is based on OECD - OECD - Test No. 421, 2015.”
(…) was added to section 2.3. Experimental Design
Based on the exposed above, we request the reviewer approve the manuscript for publication as it is.
Bueno, A., Sinzato, Y. K., Volpato, G. T., Gallego, F. Q., Perecin, F., Rodrigues, T., & Damasceno, D. C. (2020). Severity of prepregnancy diabetes on the fetal malformations and viability associated with early embryos in rats. Biology of Reproduction, 103(5), 938-950.
Hayashi, M., Morita, T., Kodama, Y., Sofuni, T., & Ishidate Jr, M. (1990). The micronucleus assay with mouse peripheral blood reticulocytes using acridine orange-coated slides. Mutation Research Letters, 245(4), 245-249.
Hu, J., Qin, X., Zhang, J., Zhu, Y., Zeng, W., Lin, Y., & Liu, X. (2021). Polystyrene microplastics disturb maternal-fetal immune balance and cause reproductive toxicity in pregnant mice. Reproductive Toxicology, 106, 42-50.
Ishikawa, R. B., Vani, J. M., das Neves, S. C., Rabacow, A. P. M., Kassuya, C. A. L., Croda, J., ... & Oliveira, R. J. (2018). The safe use of Doliocarpus dentatus in the gestational period: Absence of changes in maternal reproductive performance, embryo-fetal development and DNA integrity. Journal of ethnopharmacology, 217, 1-6.
Kamali, M., Johari, H., & Hami, J. (2020). Protective effect of flax seed on brain teratogenicity induced by lamotrigine in rat fetuses. Folia Medica, 62(2), 372-377.
Koch, E., Nöldner, M., & Leuschner, J. (2013). Reproductive and developmental toxicity of the Ginkgo biloba special extract EGb 761® in mice. Phytomedicine, 21(1), 90-97.
Lu, K., Wang, F., Ma, B., Cao, W., Guo, Q., Wang, H., ... & Wang, Z. (2021). Teratogenic Toxicity Evaluation of Bladder Cancer-Specific Oncolytic Adenovirus on Mice. Current Gene Therapy, 21(2), 160-166.
OECD - Test No. 421: Reproduction/Developmental Toxicity Screening Test, 2015. doi:10.1787/9789264242692-en
Russell, W. M. S.; Burch, R. L. (1959). The principles of humane experimental technique. Methuen.
Saeed, M., Saleem, U., Anwar, F., Ahmad, B., & Anwar, A. (2020). Inhibition of valproic acid-induced prenatal developmental abnormalities with antioxidants in rats. ACS omega, 5(10), 4953-4961.
Saini, S., Nair, N., & Saini, M. R. (2013). Embryotoxic and teratogenic effects of nickel in Swiss albino mice during organogenetic period. BioMed research international, 2013.
Scheller, K., Kalmring, F., & Schubert, J. (2016). Sex distribution is a factor in teratogenically induced clefts and in the anti-teratogenic effect of thiamine in mice, but not in genetically determined cleft appearance. Journal of Cranio-Maxillofacial Surgery, 44(2), 104-109.
Stoev, S. D. (2022). Studies on teratogenic effect of ochratoxin A given via mouldy diet in mice in various sensitive periods of the pregnancy and the putative protection of phenylalanine. Toxicon, 210, 32-38.
Vani, J. M., de Carvalho Schweich-Adami, L., Auharek, S. A., Antoniolli-Silva, A. C. M. B., & Oliveira, R. J. (2021). Pyriproxyfen does not cause microcephaly or malformations in a preclinical mammalian model. Environmental Science and Pollution Research, 28(4), 4585-4593.
Vani, J. M., de Carvalho Schweich, L., de Oliveira, K. R. W., Auharek, S. A., Cunha-Laura, A. L., Antoniolli-Silva, A. C. M. B., ... & Oliveira, R. J. (2018a). Evaluation of the effects of the larvicides temephos on reproductive performance, embryofetal development and DNA integrity of Swiss mice. Pesticide biochemistry and physiology, 148, 22-27.
Vani, J. M., Monreal, M. T. F. D., Auharek, S. A., Cunha-Laura, A. L., de Arruda, E. J., Lima, A. R., ... & Oliveira, R. J. (2018b). The mixture of cashew nut shell liquid and castor oil results in an efficient larvicide against Aedes aegypti that does not alter embryo-fetal development, reproductive performance or DNA integrity. Plos one, 13(3), e0193509.
- Another limitation of the study is the absence of a positive control for the micronucleus assay which is needed to confirm the ability of the test system to detect substances that induce chromosomal damage. Reference is also not made to historical vehicle control data for the laboratory which would support the conclusion that the higher frequency of micronuclei at the high dose compared with controls (Figure 1A) is not suggestive of a positive response. This should be discussed.
Response:
We thank the reviewer's comments. Our research group has expertise in ​​genotoxicity and publications have been frequent since 2006. Therefore, this technique is well standardized and has been in our domain for over 16 years. We do not present a historical vehicle control because we believe that experiments carried out at different times have individualities that are inherent to the animal colony tested. We have already used animals provided by different animal facilities and they show variations in the basal frequencies of DNA damage (micronuclei, for example). We believe that this is due to intraspecific differences (since we do not work with isogenic animals) and to the different conditions that the animals are subjected to, even though they are in a facility where conditions are controlled. We emphasize that we receive all batches of animals with sanity control and, therefore, they are healthy animals and free of pathogens. Furthermore, even when the animals come from the same source, if the tests are carried out at different times, we noticed differences in the basal frequencies of DNA damage. Once again, we infer that these occur by the same hypotheses already presented. Thus, we believe that historical vehicle control is not the best benchmark. Therefore, we chose to carry out a negative control group for each experiment that is conducted even if they are conducted simultaneously. The negative control group uses the highest dose of test product diluent. Therefore, the indicated fragility does not exist, that is, the frequency of micronuclei at the highest dose is not suggestive of a positive response because (I) the negative control received exactly the same dose as Tween 80 (1%) than the group that received the higher dose of extract; and (II) the frequency of the negative control and the highest dose of the extract are statistically identical. Therefore, the concern raised by the reviewer is not supported by the present design and results found. We also emphasize that the scale of the micronucleus graph is 2 units. If the highest mean plus the standard deviation is observed, the value is close to 5 micronuclei. Therefore, the frequency is very low, which does not indicate and does not allow to suggest a genotoxic effect for the extract. But, to meet the reviewer's suggestion, we added to the discussion a reference using the same strain of mice that indicate that this basal frequency is observed in other experiments (Ishikawa et al., 2018; Vani et al., 2018a; Vani et al., 2018b; Vani et al., 2021).
(...)
“The frequency of chromosomal damage, observed even for the highest dose of EECc, is within what is recommended by the literature as basal damage. According to Ishikawa et al. (2018), the average frequency of micronuclei for the negative control, treated with 1% Tween 80 (the same diluent and the same concentration used in this study) can reach 7.1 ± 1.03 micronuclei. Furthermore, Vani et al., (2018a; 2018b; 2021) reported the basal frequency of micronuclei for pregnant female Swiss mice of up to 8.90 ± 1.35, 5.20 ±1.31 and 8.90 ± 1.35, respectively.
In addition, the animals that are used for genotoxicity tests are the same animals used for teratogenesis tests. Thus, using a positive control (Cyclosphofamide (20mg/Kg) as suggested by Oliviera et al. (2009)) would lead to the description of a large number of malformations in the tables, which would make it difficult to understand the study. Since the magnitude of malformations found in the positive controls would in no way resemble what is induced by a medicinal plant with a low capacity to induce teratogenesis. Furthermore, we would not respect the 3Rs Principle discussed earlier. We can say that the CEUA would not authorize the use of 10 more pregnant females that would give approximately 80-100 more pups. The removal of this group is essential to meet the 3Rs Principle and saves the use and life of at least 100 experimental animals. In view of the above, we are not willing to change the experimental design.
We also emphasize that we are not aware of a teratogenic drug that can be used for treatment throughout the gestational period (1st to 18th gestational day). Thus, any other design that we used would not fit the proposed model. If we were to use a group to be treated with a drug known to be genotoxic only for positive control of the micronucleus test, we would certainly not be, again, respecting the 3Rs Principle. Thus, we assume that the technique has always been well standardized in our lab and we assume that genetic damage (micronuclei) should be considered whenever they are statistically different from the negative control. In addition, the appearance of micronuclei (even at low frequency) proves that the technique can confirm the ability of the test system to detect substances that induce chromosomal
We highlight that we published in the last few year the same approach in high impact scientific jornals (Vani et al., 2021; Ishikawa et al., 2018; Vani et al., 2018a; Vani et al., 2018b; Pessatto et al., 2017; de David et al., 2014; Gonçalves et al., 2013). We request the reviewer consider this manuscript as it is.
de David, N., de Oliveira Mauro, M., Gonçalves, C. A., Pesarini, J. R., Strapasson, R. L. B., Kassuya, C. A. L., ... & Oliveira, R. J. (2014). Gochnatia polymorpha ssp. floccosa: bioprospecting of an anti-inflammatory phytotherapy for use during pregnancy. Journal of ethnopharmacology, 154(2), 370-379.
Gonçalves, C. A., Siqueira, J. M., Carollo, C. A., de Oliveira Mauro, M., de Davi, N., Cunha-Laura, A. L., ... & Oliveira, R. J. (2013). Gestational exposure to Byrsonima verbascifolia: Teratogenicity, mutagenicity and immunomodulation evaluation in female Swiss mice. Journal of Ethnopharmacology, 150(3), 843-850.
Ishikawa, R. B., Vani, J. M., das Neves, S. C., Rabacow, A. P. M., Kassuya, C. A. L., Croda, J., ... & Oliveira, R. J. (2018). The safe use of Doliocarpus dentatus in the gestational period: Absence of changes in maternal reproductive performance, embryo-fetal development and DNA integrity. Journal of ethnopharmacology, 217, 1-6.
Pessatto, L. R., Auharek, S. A., Gonçalves, C. A., de David, N., Monreal, A. C. D., Kassuya, C. A. L., ... & Oliveira, R. J. (2017). Effects of dichloromethane and butanol fractions of Gochnatia polymorpha subsp. floccosa in maternal reproductive outcome, embryo-fetal development and DNA integrity in mice. Journal of ethnopharmacology, 200, 205-208.
Vani, J. M., de Carvalho Schweich-Adami, L., Auharek, S. A., Antoniolli-Silva, A. C. M. B., & Oliveira, R. J. (2021). Pyriproxyfen does not cause microcephaly or malformations in a preclinical mammalian model. Environmental Science and Pollution Research, 28(4), 4585-4593.
Vani, J. M., de Carvalho Schweich, L., de Oliveira, K. R. W., Auharek, S. A., Cunha-Laura, A. L., Antoniolli-Silva, A. C. M. B., ... & Oliveira, R. J. (2018a). Evaluation of the effects of the larvicides temephos on reproductive performance, embryofetal development and DNA integrity of Swiss mice. Pesticide biochemistry and physiology, 148, 22-27.
Vani, J. M., Monreal, M. T. F. D., Auharek, S. A., Cunha-Laura, A. L., de Arruda, E. J., Lima, A. R., ... & Oliveira, R. J. (2018b). The mixture of cashew nut shell liquid and castor oil results in an efficient larvicide against Aedes aegypti that does not alter embryo-fetal development, reproductive performance or DNA integrity. Plos one, 13(3), e0193509.
- Some more specific comments are below.
- a) Abstract
The English language used in the abstract needs to be clarified. For example:
Lines 28-30: 'Given the above, to evaluate the effects of ethanolic extract of EEGc on reproductive performance, embryo development and chromosome stability.' Could change to something along the lines of 'Given the above, a study was undertaken to evaluate...'
Lines 35-37: 'The evaluation of the absolute and relative organ weights did not vary between the different experimental groups, it also did not change the number of implants...' It is not the evaluation that didn't change, it is the test item that didn't affect these parameters. Could possibly reword to say that there were no significant differences in these compared with controls..
Lines 39-40: 'The frequency of chromosomal damage did not change by EEGc in the frequency of micronuclei.' Could change to e.g. 'the frequency of chromosomal damage following treatment with EEGc was similar to controls', or was not affected by treatment.
- b) Introduction
It would be helpful to be clearer about how much information is available on the safety of EEGc to date, or that there are few data on this. A recent paper on safety is not included and could be cited here and/or in the discussion: Preclinical safety evaluation of the ethanolic extract from the aerial parts of Gomphrena celosioides Mart. in rodents - PubMed (nih.gov)
Response:
We thank the reviewer for the comment. The requested information was added:
Not much information is known about the toxicological potential of G. celosioides. However, according to Marcorini et al. (2022), a single dose of 2000mg/kg does not induce clinical signs of toxicity in Wistar rats, which suggests that the LD50 is higher at this dose. Subacute toxicity, repeated doses, showed that 75, 150 and 300mg/Kg did not induce adverse clinical signs or damage to target tissues. According to these authors, according to the Globally Harmonized System of classification, the EEGC dosages can be in Category 5 which is at least toxic or non-toxic. No reports of toxicity in pregnant females were found in the literature consulted.
- c) Materials and Methods
Lines 121-123: the OECD references to support the high dose selected are not appropriate: neither of these studies (acute toxicity study and reproduction/developmental toxicity screening study) were conducted here. Further, the limit dose suggested in OECD TG 421 is 1000 mg/kg bw/day, not 2000 mg/kg bw/day.
Response:
We thank the reviewer for the comment. The lines were rewritten to fit with the reviewer’s request:
(…) The 100 mg/kg dose was chosen from the effective dose described by Vasconcelos et al. (2017). From this dose, the dose of 10x higher (1,000 mg/kg) than that with the desired activity (ANVISA, 2010; OECD - Test nº 421, 2015), with modifications, and the limit dose of 2,000 mg/kg was defined according to the acute oral toxicity guidelines (OECD - Test nº 423).
- d) Section 2.7: more clarity is needed on how the micronucleus test was performed: were the blood samples collected from the pregnant females or a satellite group? Were all 10/group examined? What cells were scored for micronuclei? OECD TG 474 recommends scoring at least 4000 immature erythrocytes. What were the criteria for acceptability of the test and for determining a positive or negative response? One would expect reference to historical control data as well as the use of a positive control to demonstrate validity of the test system.
Response:
We thanks the reviewer's comments. We do not refer to any satellite group and the experimental design presents only one experimental group for each treatment. Therefore, blood was collected from the same females that were used for the teratogenesis study. All animals from all groups were examined. Micronuclei were analyzed in pheripheral reticulocytes (this information and the original reference were inserted into the methodology). We do not rely on OECD TG 474. We rely on the classic work of Hayashi et al. (1990) who described the variation of the micronucleus technique so that different samples could be evaluated from the same animal using acridine orange dye. In addition, modifications were incorporated according to Carvalho et al. (2015) which is one of our papers that has the micronucleated cells shown in photos. It is also noteworthy that Hayashi et al. (1990) and Carvalho et al. (2015) analyzed 1000 reticulocytes/animal. Thus, we added the reference by Navarro et al. (2104) who analyzed 2000 reticulocytes/animal. We chose this higher number because it is a frequent number in the literature and it is how our group has contributed to the growth of the research field (Stein et al., 2022; Moreira et al., 2021; Vani et al., 2021; Vasconcelos et al., 2021; Vasconcelos et al. al., 2020; Oliveira et al., 2019; Oliveira et al. 2018; Vani et al. 2018a; Vani et al., 2018b). In view of the above, we request that the reviewer consider the manuscript in its present form.
Carvalho, P. C., Santos, E. A., Schneider, B. U. C., Matuo, R., Pesarini, J. R., Cunha-Laura, A. L., ... & Oliveira, R. J. (2015). Diaryl sulfide analogs of combretastatin A-4: Toxicogenetic, immunomodulatory and apoptotic evaluations and prospects for use as a new chemotherapeutic drug. Environmental toxicology and pharmacology, 40(3), 715-721.
Hayashi, M., Morita, T., Kodama, Y., Sofuni, T., & Ishidate Jr, M. (1990). The micronucleus assay with mouse peripheral blood reticulocytes using acridine orange-coated slides. Mutation Research Letters, 245(4), 245-249.
Moreira, F. M. F., Radai, J. A. S., de Souza, V. V., Berno, C. R., de Araújo, F. H. S., Andrade-Silva, M., ... & Croda, J. (2021). Toxicological analysis and efficacy of 2-phenylchromone on mycobacteria viability and inflammatory response induced by Mycobacterium bovis. Phytomedicine Plus, 1(4), 100117.
Navarro, S. D., Beatriz, A., Meza, A., Pesarini, J. R., da Silva Gomes, R., Karaziack, C. B., ... & Oliveira, R. J. (2014). A new synthetic resorcinolic lipid 3-Heptyl-3, 4, 6-trimethoxy-3H-isobenzofuran-1-one: Evaluation of toxicology and ability to potentiate the mutagenic and apoptotic effects of cyclophosphamide. European journal of medicinal chemistry, 75, 132-142.
Oliveira, R. J., da Cruz Leite Santos, N., Pesarini, J. R., de Oliveira, B. C., Berno, C. R., de Araújo, F. H. S., ... & da Silva Gomes, R. (2018). Assessment of genetic integrity, splenic phagocytosis and cell death potential of (Z)-4-((1, 5-dimethyl-3-oxo-2-phenyl-2, 3dihydro-1H-pyrazol-4-yl) amino)-4-oxobut-2-enoic acid and its effect when combined with commercial chemotherapeutics. Genetics and Molecular Biology, 41, 154-166.
Oliveira, R. J., Pereira, F. P. A. N., Silveira, I. O. M. F. D., Lima, R. V. D., Berno, C. R., Pesarini, J. R., ... & Gomes, R. D. S. (2019). Assessment of the toxicogenic effects and cell death potential of the ester (Z)-methyl 4-((1, 5-dimethyl-3-oxo-2-phenyl-2, 3-dihydro-1H-pyrazol-4-yl) amino)-4-oxobut-2-anoate in combination with cisplatin, cyclophosphamide and doxorubicin. Genetics and Molecular Biology, 42, 399-410.
Stein, J., Jorge, B. C., Reis, A. C. C., Radai, J. A. S., da Silva Moreira, S., Fraga, T. L., ... & Arena, A. C. (2022). Evaluation of the safety of ethanolic extract from Piper amalago L.(Piperaceae) leaves in vivo: Subacute toxicity and genotoxicity studies. Regulatory Toxicology and Pharmacology, 129, 105118.
Vasconcelos, N. G., Vaz, M. S. M., Radai, J. A. S., Kassuya, C. A. L., Formagio, A. S. N., Graciani, F. S., ... & Simionatto, S. (2020). Antimicrobial activity of plant extracts against carbapenem-producing Klebsiella pneumoniae and in vivo toxicological assessment. Journal of Toxicology and Environmental Health, Part A, 83(23-24), 719-729.
Vani, J. M., de Carvalho Schweich-Adami, L., Auharek, S. A., Antoniolli-Silva, A. C. M. B., & Oliveira, R. J. (2021). Pyriproxyfen does not cause microcephaly or malformations in a preclinical mammalian model. Environmental Science and Pollution Research, 28(4), 4585-4593.
Vani, J. M., de Carvalho Schweich, L., de Oliveira, K. R. W., Auharek, S. A., Cunha-Laura, A. L., Antoniolli-Silva, A. C. M. B., ... & Oliveira, R. J. (2018a). Evaluation of the effects of the larvicides temephos on reproductive performance, embryofetal development and DNA integrity of Swiss mice. Pesticide biochemistry and physiology, 148, 22-27.
Vani, J. M., Monreal, M. T. F. D., Auharek, S. A., Cunha-Laura, A. L., de Arruda, E. J., Lima, A. R., ... & Oliveira, R. J. (2018b). The mixture of cashew nut shell liquid and castor oil results in an efficient larvicide against Aedes aegypti that does not alter embryo-fetal development, reproductive performance or DNA integrity. Plos one, 13(3), e0193509.
- e) Statistics: the appropriate statistical unit for evaluating developmental toxicity studies is the litter. It is unclear if this was used for this study.
Response:
We thank the reviewer for the comment. We chanfed the sentence and the requested information was incorporated:
(…) The qualitative data and the frequencies obtained, as recommended by the specialized literature, had the offspring used as the base-unit. (Haseman; Hogan, 1975; Manson; Zenick; Costlow, 1982).
Manson, J. M., Zenick, H., & Costlow, R. D. (1982). Teratology test methods for laboratory animals. Principles and methods of toxicology, 141-184.
Haseman, J. K., & Hogan, M. D. (1975). Selection of the experimental unit in teratology studies. Teratology, 12(2), 165-171.
- f) Results: Lines 228-229: refers to a significant decrease in placental efficiency at the low and high dose. Lines 266-267: refers to increased frequency of abnormal sternum and rotation of the fibula at the low dose only. However, a clear dose-response was not observed for any of these changes, have the authors considered that these might be random statistical differences rather than treatment-related? Reference to historical control data would be helpful to interpret these results.
Response:
We appreciate the reviewer's comments. Yes, we considered that these might be random statistical differences rather than treatment-related. We disagree with the need for a historical control data to discuss the data. We can do this discussion based on the negative control results in association with data from the literature. We even cited in our discussion articles that reinforce our hypothesis that these results have no biological relevance and/or are variants of normality. We are comfortable with this way of presenting the data and ask that the reviewer consider the manuscript in its present form. Perhaps in the near future we will be able to build a new manuscript with different negative controls and a naïve group to present what would be the history of our mouse colony. However, at this point, we believe that the manuscript has enough data for the present discussion and there is no need to compile all the data we have from previous experiments. What prevents us from compiling this compilation are all the arguments previously presented for the micronucleus assay and we also emphasize that the studies were carried out by different researchers. Therefore, analyzes performed by more than one observer can induce biases that we do not want. Thus, we reinforce the need for the reviewer to consider the present form of the manuscript. We also ask the reviewer to point out in the literature in the area a paper on micronucleus and another on teratogenesis that make use of historical control data so we can study the situation more closely and perhaps adopt the suggestion in future research papers.
Lines 248-250: it is stated that treatment had no effect on the frequency of malformations (Table 4). However, several significant differences are shown in Table 4. Again these may not be treatment-related as they did not show a clear dose-response.
Response:
The text has been changed and we have added the word total to indicate that the total frequency analysis did not indicate a significant increase in the frequency of malformations although the individualized malformation analysis indicated a significant increase in unilateral limb hyperflexion. We agree that this effect is due to treatment, and we discussed this in the discussion section. We reported in the discussion that despite this effect such malformations are of lower risk. Therefore, we ask attention to this point of the discussion and request that the reviewer consider the manuscript in its present form.
- g) Discussion
Reference is made to changes in placental efficiency and malformations, but as noted above more consideration should be given as to whether these are treatment-related or due to chance.
Response:
We believe the finding is due to treatment. However, this isolated data has no biological relevance. We have modified the text to make our position clearer.
(…)
The reduction in placental efficiency, even being an effect of the treatment, did not determine variation in the adequacy of weight to the age of pregnant women or in the adequacy of weight to gestational age.
- h) Discussion of limitations of the study would be useful, e.g. absence of a positive control for the micronucleus assay, no reference to historical control data etc.
Response:
We appreciate the reviewer's comments. All the justifications for these questions were previously presented and even changes in the manuscript were made to meet the requests. Therefore, we ask the reviewer to consider the manuscript in its present form.
- i) Discussion of what further research might be needed to confirm the safety of medicinal use of EEGc would be useful.
Response:
A paragraph was added.
(…)
Other experimental designs are available in the literature and can complement the data produced by this study, which is novel and pioneer in presenting the effects of G. celosioideson maternal performance and embryofetal development. The specific protocols of the OECD - Test No. 421 (2016), OECD - Test No. 414 (2017) and OECD - Test 443 (2018) guidelines should be considered in future studies.
Thus, I think I have complied with the points raised by both reviewers.
I hope that the revised manuscript can be published as is without further delay.
With all good wishes,
Roberto Gomes

Round 2
Reviewer 2 Report
I thank the authors for their responses and changes and consider that the paper could be accepted for publication, however I would note the following points and suggest that addressing these should be considered where relevant:
Regarding control data for the genotoxicity assay: the authors' rationale for not including a positive control given the study design makes sense; it would be useful for this to be explained either in the methods or discussion section to address this as the absence of a positive control may otherwise be interpreted as a limitation.
Statistically significant changes in malformation frequency that did not show a dose response: suggest the authors at least acknowledge in the discussion that these may not necessarily be treatment-related given the absence of a dose-response.
The authors ask for references in the literature where historical control data are referred to in micronucleus or teratotogenicity studies. Without conducting a literature search I am not able to provide examples within the publishers' requested time-frame, however the following review paper on genotoxicity of titanium dioxide highlights the need for historical control data: A weight of evidence review of the genotoxicity of titanium dioxide (TiOâ‚‚) - ScienceDirect Furthermore, the OECD test guidelines for the in vivo mammalian micronucleus test (TG 474), prenatal developmental toxicity study (TG 414) and reproduction/developmental toxicity screening test (TG 421) all highlight the value of historical control data and such data are regularly provided in reports of studies conducted in accordance with these test guidelines (note many such studies that I have reviewed are unpublished as they are proprietary study reports).
Author Response
Response to reviewer 2:
I thank the authors for their responses and changes and consider that the paper could be accepted for publication, however I would note the following points and suggest that addressing these should be considered where relevant:
- Regarding control data for the genotoxicity assay: the authors' rationale for not including a positive control given the study design makes sense; it would be useful for this to be explained either in the methods or discussion section to address this as the absence of a positive control may otherwise be interpreted as a limitation.
Response:
We thank the reviewer for the suggestions. As requested, we have added a new paragraph explaining either in the methods or discussion section to address this as the absence of a positive control may otherwise be interpreted as a limitation:
The same animals were used in the teratogenicity and genotoxicity tests. However, there is no positive control for these two assays simultaneously, and that can still follow the same proposed experimental design, ie, be administered for 18 consecutive days. Thus, we chose not to use a positive control group for this study since this could negatively impact the treatment schedule. Moreover, this study was delineated to follow the philosophy of the 3Rs (Replacement, Reduction and Refinement) proposed by Russell and Burch (1959). Another fact that prevents us from performing the positive control group is that products known to be teratogenic, such as Cyclophosphamide (20mg/Kg) (Oliveira et al., 2009) would lead to the description of a large number of malformations in the tables, which would make it difficult to understand the results. This would happen because the magnitude of the malformations found in positive controls would in no way resemble what is induced by a medicinal plant with a low capacity to induce teratogenesis.
- Statistically significant changes in malformation frequency that did not show a dose response: suggest the authors at least acknowledge in the discussion that these may not necessarily be treatment-related given the absence of a dose-response.
Response:
We thank the reviewer for the suggestions. As requested, we have added a new paragraph acknowledging in the discussion that these may not necessarily be treatment-related given the absence of a dose-response:
Another fact to be highlighted is that the malformations that presented statistically significant differences did not present a dose-response relationship, that is, what was expected was that with the increase in the cEEG dose, there will have also an increase in the frequency or severity of malformations. However, this fact was not observed. Thus, the effect may not be caused by the administration of the cEEG. In this case, the hypothesis is that these are in fact variants of normality.
- The authors ask for references in the literature where historical control data are referred to in micronucleus or teratotogenicity studies. Without conducting a literature search I am not able to provide examples within the publishers' requested time-frame, however the following review paper on genotoxicity of titanium dioxide highlights the need for historical control data: A weight of evidence review of the genotoxicity of titanium dioxide (TiOâ‚‚) - ScienceDirect Furthermore, the OECD test guidelines for the in vivo mammalian micronucleus test (TG 474), prenatal developmental toxicity study (TG 414) and reproduction/developmental toxicity screening test (TG 421) all highlight the value of historical control data and such data are regularly provided in reports of studies conducted in accordance with these test guidelines (note many such studies that I have reviewed are unpublished as they are proprietary study reports).
Response:
We thank the reviewer for the kind reply. We will keep this in mind to use it when future experiments were delineated.
Thus, I think I have complied with the points raised by both reviewers.
I hope that the revised manuscript can be published as is without further delay.
With all good wishes,
Roberto Gomes
